# A Data-Driven Space-Time-Parameter Reduced-Order Model with Manifold Learning for Coupled Problems: Application to Deformable Capsules Flowing in Microchannels

**DOI:** 10.3390/e23091193

**Published:** 2021-09-09

**Authors:** Toufik Boubehziz, Carlos Quesada-Granja, Claire Dupont, Pierre Villon, Florian De Vuyst, Anne-Virginie Salsac

**Affiliations:** 1Biomechanics and Bioengineering Laboratory (UMR CNRS 7338), Université de Technologie de Compiègne CNRS, Alliance Sorbonne Université, 60203 Compiègne, France; toufik.boubehziz@utc.fr (T.B.); quesadagranja@gmail.com (C.Q.-G.); claire.dupont@utc.fr (C.D.); a.salsac@utc.fr (A.-V.S.); 2Laboratoire Roberval, Université de Technologie de Compiègne, Alliance Sorbonne Université, 60203 Compiègne, France; pierre.villon@utc.fr; 3Laboratoire de Mathématiques Appliquées de Compiègne (EA 2222), Université de Technologie de Compiègne, Alliance Sorbonne Université, 60203 Compiègne, France

**Keywords:** data-driven model, model order reduction, proper orthogonal decomposition, manifold learning, diffuse approximation, microcapsule suspension, Hausdorff distance

## Abstract

An innovative data-driven model-order reduction technique is proposed to model dilute micrometric or nanometric suspensions of microcapsules, i.e., microdrops protected in a thin hyperelastic membrane, which are used in Healthcare as innovative drug vehicles. We consider a microcapsule flowing in a similar-size microfluidic channel and vary systematically the governing parameter, namely the capillary number, ratio of the viscous to elastic forces, and the confinement ratio, ratio of the capsule to tube size. The resulting space-time-parameter problem is solved using two global POD reduced bases, determined in the offline stage for the space and parameter variables, respectively. A suitable low-order spatial reduced basis is then computed in the online stage for any new parameter instance. The time evolution of the capsule dynamics is achieved by identifying the nonlinear low-order manifold of the reduced variables; for that, a point cloud of reduced data is computed and a diffuse approximation method is used. Numerical comparisons between the full-order fluid-structure interaction model and the reduced-order one confirm both accuracy and stability of the reduction technique over the whole admissible parameter domain. We believe that such an approach can be applied to a broad range of coupled problems especially involving quasistatic models of structural mechanics.

## 1. Introduction

Numerical modeling and simulation today appear to be an indispensable science to analyze physics-coupled problems (e.g., micrometric and nanometric suspensions), but also for innovative design and optimization of complex three-dimensional systems in engineering and industry (health, automotive, aircraft, etc.). Although one can nowadays find robust and accurate open-source or commercial codes for the simulation of multiphysics systems, it is still hard to use them in the context of robust or optimal design because of the prohibitive computational time that does not match with engineering production horizons. In order to accelerate computations, one can make use of parallel HPC (High Performance Computing) facilities, but this can become financially costly. For most applications, even with HPC facilities, the evaluation of the solutions takes days or weeks for three-dimensional multi-coupled problems.

Alternatively, a current tendency is to use machine learning or artificial intelligence tools to capitalize knowledge stored into data and use it for future case studies. It leads to less redundant or useless computations, while the database of results continues to grow with more and more relevant information contents. With machine learning, one can expect to explore the design spaces in an easier, faster and more efficient way. However, one usually needs expertise to design and train artificial neural networks (ANN) correctly. For high-dimensional data, the training stage may require large computational resources and issues of quality of data may also be raised. Machine learning can be extended to time dependent problems and dynamical system using e.g., recurrent neural networks [1].

However, for particular use cases like physical systems, machine learning algorithms are designed to return three-dimensional spatial fields. This means that the outputs of the networks are high-dimensional vectors, which may induce training convergence and accuracy issues. Over the last two years, one could observe the rise of so-called Physics-informed neural networks (PINN) where the artificial neural networks are trained from a loss function that includes physical information (like partial differential equations), see e.g., [2].

Another class of ’machine learning’ methods are the data-driven model order reduction (MOR) techniques, that use data generated from a (time-consuming) high-fidelity solver, also called the full-order model (FOM) [3,4]. Data-driven and non-intrusive reduced-order models (ROM) can be seen as supervised ANN [5,6]. For parametrized partial differential problems, ROMs usually perform a dimensionality reduction by means of suitable reduced bases. This can be achieved via different approaches such as the Proper Orthogonal Decomposition (POD) [7,8,9], piecewise tangential interpolation [10], Proper Generalized Decompositions (PGD) [11,12,13], Empirical Interpolation Methods (EIM) [14,15,16] or via different greedy procedures [17]. Then one has to find the manifold that maps the parameters to the coefficients of linear combination of the reduced basis functions. This can also be achieved in a supervised way, by means of universal approximation techniques like diffuse approximation [18] for example. Using ROMs may lead to substantial speedups as compared to FOM, from say 10 up to 10,000. One can even imagine real-time computations in some cases [19].

The ’ultimate’ case is that of space-time-parameter problems involving spatial fields, timeline and design variables. This is of course of industrial importance, but still an issue and a current active field of research (see [20] for example). For such problems, the data are generally organized in data cubes (Figure 1). In this paper, a data-driven reduced-order modeling approach is proposed for space-time-parameter mechanical problems involving an equation of kinematics and a quasi-static law of equilibrium. As particular application, the physical problem that is addressed is the dynamics of dilute suspensions of micrometric capsules in microfluidic channels. Microcapsules can be used in Healthcare as innovative drug transportation vehicles into blood vessels and are expected to deliver drugs at identified targets [21,22]. They are composed of an elastic membrane protecting a liquid inner core and are used in suspension in another liquid. Testing them in microfluidic environments offers great potential to determine the capsule behavior and characterize the mechanical properties of the membrane [23,24,25,26,27,28,29], but also for sorting or enrichment of capsule suspensions [30,31,32,33,34]. We presently focus on the flow of a dilute suspension of initially spherical micrometric capsules in a microfluidic channel, which is a complex three-dimensional inertialess fluid-structure interaction problem that interestingly depends on only two independent design variables: the capillary number of the capsule Ca, which is a non-dimensional number that estimates the order of magnitude of the viscous forces acting on the capsule with respect to the elastic forces that build up in the membrane, and the confinement ratio a/ℓ that provides a comparison between the initial capsule diameter and the channel width.

Proper Orthogonal Decomposition has been shown to be particularly suitable to build reduced order models (ROM) of microcapsules [35], but so far no model capable of predicting capsule dynamics currently exists. The originality of the paper is to propose a ROM of the capsule-fluids interactions which provides the time-evolution of the capsule shape for any parameter values. From the capsule shape, it is indeed possible to deduce all the quantities of interest (viscous load, internal tensions within the membrane, membrane energy, etc.) in post-treatment. The ROM is inspired from the physical problem, in which the boundary condition stipulates that the fluid velocity equals the capsule membrane velocity. The correction of the capsule node position field can thus be obtained by integrating the velocity field over time. The challenge remains to correlate the position and velocity fields, which we propose to do with diffuse approximation and manifold learning [18,36,37] using the principal modes of both fields obtained by POD decomposition.

Numerical experiments will demonstrate the accuracy and efficiency of the approach, comparing reduced-order solutions to the full-order ones. We believe that the methodology proposed in this paper can be applied to a broad range of multiphysics problems such as fluid-structure interactions, structural dynamics using quasi-static structural mechanics models and related problems.

This paper is organized as follows. In Section 2, we describe the physical problem and its full order model solution. In Section 3.2, we construct parametric and spatial reduced-order modes using a set of pre-computed simulations. This allows to introduce a reduced-order model that expresses the displacement and velocity of a capsule at a selection of snapshots. In Section 3.4, we build a reduced model that corresponds to any parameter vector of Ca and a/ℓ values by estimating its corresponding principal components. Then, with the use of a Diffuse Approximation (DA) method, we adopt a data-driven manifold learning to predict the deformation of the capsule in the flow for a chosen time discretization. Finally, in Section 4, we will validate the whole computational ROM approach with a comparison to the FOM solutions.

## 2. Material and Methods

### 2.1. Problem Statement

Let us consider a spherical micrometric capsule of radius *a* freely placed in a three-dimensional microfluidic channel with square cross-section of length 2ℓ (see Figure 2). The capsule and the channel are filled with an incompressible Newtonian fluid of the same constant density ρ and dynamic viscosity μ. The capsule is enclosed by a thin hyperelastic isotropic membrane (surface shear modulus $G_{s}$, area expansion modulus $K_{s}=3G_{s}$). It is subjected to a Poiseuille flow of mean velocity *V*.

The problem is governed by two dimensionless numbers:The confinement ratio a/ℓ, ratio of the capsule to tube sizes;The capillary number $Ca=\mu V/G_{s}$, ratio of the viscous forces onto the capsule membrane to the membrane elastic forces.

The Reynolds number of the external flow is assumed to be very small (typically of order $10^{-2}$ or less), inertia being negligible because of the spatial scales involved in the problem. As far as the internal flow is concerned, its velocity is induced by the motion of the capsule membrane, which is itself entrained by the external flow: it is thus of smaller amplitude than that of the external flow. Hence, the flow in the internal $\left(\beta =in\right)$ and external $\left(\beta =ex\right)$ fluids are described by the Stokes equations:(1)$$\nabla \cdot v^{\beta }=0,\;\nabla \cdot \sigma^{\beta }=0,\;\beta =in,ex.$$
where $\sigma^{\beta }$ is stress tensor in the fluids. The Stokes equations are defined in the domains bounded by the capsule membrane for β=in and by the capsule membrane and the channel wall for β=ex. The inlet $\Gamma_{in}$ and outlet $\Gamma_{out}$ cross-sections of the channel are assumed to be far from the capsule (10ℓ in the FOM model). The reference frame $\left(O,x,y,z\right)$ is fixed on the capsule center of mass *O* at each time step. For the velocity vector field $v^{\beta }$ and the pressure field $p^{\beta }$, we consider the following boundary conditions (note that the boundary conditions include wall confinement effects, see [38] for more details):The flow perturbation induced by the capsule vanishes at $\Gamma_{in}$ and $\Gamma_{out}$:
(2)$$v^{ex}\left(x,t\right)\to v^{\infty }\left(x\right),\;\mathrm{when}\;x\in \Gamma_{in}\cup \Gamma_{out},$$
where $v^{\infty }$ is the Poiseuille flow velocity of the suspending fluid in the absence of capsule. For a square channel we have the expression in expansion form
(3)$$v^{\infty }\left(x,y\right)=\frac{\sum_{n=1,3,\ldots }^{\infty }\frac{\pi V}{n^{3}}\left[1-\frac{\cosh(n\pi x/\ell )}{\cosh(n\pi /2)}\right]\sin\left(n\pi \left(y/\ell +1/2\right)\right)}{2\left[\frac{\pi^{4}}{96}-\sum_{n=1,3,\ldots }^{\infty }\frac{\tanh(n\pi /2)}{n^{5}\pi /2}\right]}.$$Uniform pressure at $\Gamma_{in}$ and $\Gamma_{out}$:
(4)$$p^{ex}\left(x,t\right)=0,\;x\in \Gamma_{in},$$
(5)$$p^{ex}\left(x,t\right)=\Delta p\left(t\right)+\Delta p^{\infty },\;x\in \Gamma_{out},$$
where $\Delta p^{\infty }$ is the undisturbed suspending pressure drop in the absence of capsule and Δp is the additional pressure drop due to the capsule.No slip boundary conditions on the channel wall *W*:
(6)$$v^{ex}\left(x,t\right)=0,\;\mathrm{for}\;x\in W$$No slip boundary conditions on the capsule membrane *C*:
(7)$$v^{in}\left(x,t\right)=v^{ex}\left(x,t\right)=\frac{\partial }{\partial t}x\left(X,t\right),\;\mathrm{for}\;x\in C$$
where $\frac{\partial }{\partial t}x\left(X,t\right)$ is the membrane velocity at position x at time *t*, and X is the reference position vector of the capsule membrane.The normal loading continuity indicates that the load q on the membrane is due to the viscous traction jump
(8)$$\left[\sigma^{ex}\left(x\right)-\sigma^{in}\left(x\right)\right]\cdot n=q,\;\mathrm{for}\;\;x\in C$$
where n is the outward unit normal vector.

As the membrane thickness is negligibly small compared to the capsule dimensions, the membrane can be considered as a hyperelastic surface devoid of bending stiffness. The in-plane deformation is then measured by the principal extension ratios $\lambda_{1}$ and $\lambda_{2}$, that measure the in-plane deformation. Owing to the combined effects of hydrodynamic forces, boundary confinement, and membrane deformability, the capsule can be highly deformed. Consequently, the choice of membrane constitutive law is important. We consider the Neo-Hookean (NH) constitutive law that models the membrane as an infinitely thin sheet of a three-dimensional isotropic and incompressible material. It was indeed shown to adequately model microcapsules with a cross-linked proteic membrane [23,24,39]. The principal Cauchy in-plane tensions $\tau_{i}$ (i=1,2) (forces per unit arc length of deformed surface curves) can be expressed as a function of the principal extension ratios:(9)$$\tau_{1}=\frac{G_{s}}{\lambda_{1}\lambda_{2}}\left[\lambda_{1}^{2}-\frac{1}{{\left(\lambda_{1}\lambda_{2}\right)}^{2}}\right],\;\tau_{2}=\frac{G_{s}}{\lambda_{1}\lambda_{2}}\left[\lambda_{2}^{2}-\frac{1}{{\left(\lambda_{1}\lambda_{2}\right)}^{2}}\right].$$

### 2.2. Discrete Full Order Model (FOM)

The Fluid-Structure Interaction (FSI) problem is numerically modeled by coupling the Boundary Integral Method (BIM) that solves the fluid Equations (Equation 2)–(Equation 8) with the Finite Element Method (FEM) that solves the membrane mechanical problem [38,40] using the Caps3D in-house code. The unknowns are the discrete displacement field {u} and the discrete velocity field {v} at the nodes of the membrane mesh. The equation of kinematics states that $\frac{d}{dt}\left\{u\right\}=\left\{v\right\}$. The forces exerted onto the membrane are computed by the FEM. The deformation of the membrane is computed from the velocity vector field obtained at the membrane nodes by solving the Stokes equations with the BIM, leading to a nonlinear relation written in abstract form {v}={N}({u}). For a given parameter vector $\theta ={\left(\theta_{1},\theta_{2}\right)}^{T}$, where $\theta_{1}=Ca$ and $\theta_{2}=a/\ell$, the time-continuous semi-discrete FSI scheme reads in abstract form
$$\begin{array}{cc} \; & \frac{d}{dt}\left\{u\right\}\left(t\right)=\left\{v\right\}\left(t\right),\\ \; & \left\{v\right\}\left(t\right)=\left\{N\right\}\left(\left\{u\right\}\left(t\right),\theta \right),\;t\in \left(0,T_{f}\right],\\ \; & \left\{u\}(0)=\{0\},\;\{v\}(t)=\{N\}(\{0\},\theta \right) \end{array}$$
where {u}(t) and {v}(t) represent the discrete FE displacement and velocity fields at continuous time *t*, and $T_{f}$ is the final time. For time discretization, either a forward Euler scheme or a second order Runge-Kutta scheme is used with a suitable constant time step δt>0. The Euler scheme reads
$$\begin{array}{cc} \; & \left\{u^{i+1}\right\}=\left\{u^{i}\right\}+\delta t\;\left\{v^{i}\right\},\\ \; & \left\{v^{i+1}\right\}=\left\{N\right\}\left(\left\{u^{i+1}\right\},\theta \right),\\ \; & \left\{u^{0}\right\}=\left\{0\right\},\;\left\{v^{0}\right\}=\left\{N\right\}\left(\left\{0\right\},\theta \right) \end{array}$$
where $\left\{u^{i}\right\}$ and $\left\{v^{i}\right\}$ represent the discrete FE displacement and velocity fields at discrete time $t^{i,\delta }=i\;\delta t\leq T_{f}$. For second-order accuracy in time, a Runge-Kutta Ralston scheme is used:$$\begin{array}{cc} \; & \left\{{\hat{u}}^{i+2/3}\right\}=\left\{u^{i}\right\}+\frac{2}{3}\delta t\;\left\{v^{i}\right\},\\ \; & \left\{{\hat{v}}^{i+2/3}\right\}=\left\{N\right\}\left(\left\{{\hat{u}}^{i+2/3}\right\},\theta \right),\\ \; & \left\{u^{i+1}\right\}=\left\{u^{i}\right\}+\frac{\delta t}{4}\;\left(\left\{v^{i}\right\}+3\left\{{\hat{v}}^{i+2/3}\right\}\right),\\ \; & \left\{v^{i+1}\right\}=\left\{N\right\}\left(\left\{u^{i+1}\right\},\theta \right),\\ \; & \left\{u^{0}\right\}=\left\{0\right\},\;\left\{v^{0}\right\}=\left\{N\right\}\left(\left\{0\right\},\theta \right). \end{array}$$Because of the explicit nature of the numerical schemes for the equation of kinematics, the time step is subject to a Courant-Friedrichs-Lewy (CFL)-like stability condition
(10)$$\dot{\gamma }\;\delta t<C\;\frac{\Delta h_{C}}{\ell }\;Ca,$$
where $\dot{\gamma }=V/\ell$, C>0 is a constant and $\Delta h_{C}$ is the typical mesh size (see [40]). In practice, we first use small time steps, and tune them to be big enough but not too close to the stability boundary. This process does not take too much time.

### 2.3. Design of Experiment, Database of FOM Results

Simulations of the FOM problem have been run varying the governing parameters in the range [0;0.2] for the capillary number Ca and [0.75;1.2] for the confinement ratio a/ℓ. For a/ℓ≥0.95, the capsule is initially pre-deformed into an ellipsoid of semi-minor axis equal to 0.9. This pre-deformation does not have any impact on the steady-state capsule shape and is enough to avoid contacts between the capsule membrane and the channel wall. The resulting numerical database, composed of $N_{c}=118$ (Ca, a/l) samples (Figure 3), contains the time-evolution of the three-dimensional position (or displacements) vectors of the capsule membrane nodes. Only the configurations for which a steady-state shape has been reached are retained. No steady state is found above the dotted red line of Figure 3, the microcapsules exhibiting continuous elongation owing to the strain-softening behavior of the membrane law [41].

In the ROM model, we consider the capsule positions in the laboratory reference frame (and not the reference frame centred on the capsule centre of mass as in the FOM model). The capsule thus moves along the microchannel. For data generation, we pick up time snapshot solutions at coarser discrete times $t^{i}=i\;\Delta t$, where Δt=mδt for some integer m≥1. The total number of coarse discrete times is denoted $N_{t}$. Let ${\left\{X\right\}}_{\left[n\right]}$ and ${\left\{x\right\}}_{\left[n\right]}\left(t^{i}\right)\in R^{3}$, $n=1,\ldots ,N_{x}$, be the coordinates of node number *n* of the capsule mesh in the reference configuration (at time t=0) and at discrete time $t^{i}$, $i=1,\ldots ,N_{t}$. The coordinates in the current configuration $\left\{x\right\}\left(t^{i}\right)$ are function of the (Ca, a/ℓ) parameter value denoted $\theta_{j}$, $j=1,\ldots ,N_{c}$. The database is thus stored as a datacube of 3D-space, 1D-time and 2D-parameter data. The displacement vector is then $\left\{u\right\}\left({\left\{X\right\}}_{\left[n\right]},t^{i},\theta_{j}\right)={\left\{x\right\}}_{\left[n\right]}\left(t^{i}\right)-{\left\{X\right\}}_{\left[n\right]}$. The velocity vector {v} is calculated by finite differences from the position vector. Typically, for a standard capsule FOM simulation, $N_{x}$ is of order $10^{3}$ and $N_{c}$ of order $10^{2}$. The time step Δt is chosen such that $N_{t}$ is of order $10^{2}$.

## 3. Reduced Order Model (ROM)

Reduced order modeling aims at deriving a lightweight model of low-order dimension from solutions obtained by the FOM, while trying to keep the same order of accuracy. There are many reasons for doing that. In particular, parameter exploration and sensitivity analysis are made easier because of large speedups using the ROM compared to the prohibitive FOM computational time. One can also imagine real-time parameter exploration and visualization of capsule evolution.

### 3.1. Overview

We first give a general overview of the proposed data-driven model order reduction methodology. The approach is classically made of an offline stage for the search of the principal components and POD coefficient matrices of the FOM solutions, followed by an online stage where a parameter is chosen and a low-order dynamical system is run to get the solutions.

**Offline stage**. We reduce the data dimensionality by means of a double POD basis for space and parameter variables. The displacement field is represented as
(11)$$\left\{u\right\}\left(\left\{X\right\},t,\theta \right)=\sum_{k=1}^{K_{u}^{x}}\sum_{\ell =1}^{K_{u}^{c}}A_{k\ell }\left(t\right)\;{\left\{\Phi_{u}^{r}\right\}}_{k}\;{\left(\psi_{u}\left(\theta \right)\right)}_{\ell },$$
where ${\left\{\Phi_{u}^{r}\right\}}_{k}\in R^{3N_{x}}$ are the spatial POD modes, $\psi_{u}\left(\theta \right)\;\in R^{K_{u}^{c}}$ the parameter modes and $A_{k\ell }\left(t\right)$ scalar coefficients depending on time *t*. The truncation ranks are $K_{u}^{x}$ and $K_{u}^{c}$, respectively (the ‘*x*’ superscript stands for ‘space’ and the ‘*c*’ superscript for ‘configuration’). We use a similar representation for the velocity field:
(12)$$\left\{v\right\}\left(\left\{x\right\},t,\theta \right)=\sum_{k=1}^{K_{v}^{x}}\sum_{\ell =1}^{K_{v}^{c}}B_{k\ell }\left(t\right)\;{\left\{\Phi_{v}^{r}\right\}}_{k}\;{\left(\psi_{v}\left(\theta \right)\right)}_{\ell }.$$The determination of the double POD basis is achieved by singular value decomposition (SVD) from the datacube with different rearrangements of the data in stacked matrix form. The truncation ranks $K_{u}^{x}$, $K_{u}^{c}$, $K_{v}^{x}$, $K_{v}^{c}$ are expected to be rather small while ensuring accuracy of the representations.**Online stage**. For any query parameter $\theta_{q}$ in the parameter domain:(a)Estimate the displacement field $\left\{u\right\}(\left\{X\right\},t,\theta_{q})$ from expression (Equation 11). This requires an interpolation process at $\theta =\theta_{q}$. For that, we decide to use a diffuse approximation technique [18] that can be used for any parameter space dimension;(b)From the estimated displacement field $\left\{u\right\}(\left\{x\right\},t^{i},\theta_{q})$ computed at different instants $t^{i}\in \left[0,T_{f}\right]$, compute a low-order reduced basis $\left\{\varphi^{k}\right\}\left(\theta_{q}\right)$, $k=1,\ldots ,m_{u}$ by singular value decomposition. We then get the low-order representations of both displacements and velocities:
(13)$$\begin{array}{cc} \; & \left\{u\right\}\left(\left\{X\right\},t,\theta_{q}\right)=\sum_{k=1}^{m_{u}}\alpha_{k}\left(t\right)\;\left\{\varphi^{k}\right\}\left(\theta_{q}\right), \end{array}$$
(14)$$\begin{array}{cc} \; & \left\{v\right\}\left(\left\{X\right\},t,\theta_{q}\right)=\sum_{k=1}^{m_{v}}\xi_{k}\left(t\right)\;\left\{\gamma^{k}\right\}\left(\theta_{q}\right), \end{array}$$(c)Manifold learning online stage: using diffuse approximation, we determine the low-order manifold M that links displacements and velocities in the (reduced-order) state space:
$$\xi =M_{}\left(\alpha ,\;\theta_{q}\right);$$(d)Derivation of a low-order dynamical system: we then derive a lightweight differential-algebraic dynamical system, easy to solve numerically: for $\theta =\theta_{q}$, solve
$$\begin{array}{cc} \; & \frac{d\alpha }{dt}=Q\;\xi \left(t\right),\\ \; & \xi \left(t\right)=M_{}\left(\alpha \left(t\right),\;\theta_{q}\right). \end{array}$$The high-dimensional displacement and velocity fields can then be reconstructed according to (Equation 13) and (Equation 14).

In the next section, we give all the details of the ROM methodology.

### 3.2. Offline Stage

#### 3.2.1. Global Parametric Reduced Basis (GPRB)

This first step consists in computing a parametric reduced basis in the whole parameter domain from the database of FOM results (see Section 2.3). For simplification reasons, we use the subscript ϱ that can be either *u* or *v* to express displacements and velocity respectively in the formulas.

Let $S_{u}^{i}\in M_{3N_{x},N_{c}}\left(R\right)$ be the matrix of capsule displacement fields {u} and $S_{v}^{i}\in M_{3N_{x},N_{c}}\left(R\right)$ the matrix of the velocity fields {v} at time $t^{i}$, $i=1,\ldots ,N_{t}$ (Figure 4a), considering all the configurations $\theta_{j}$ for $j=1,\ldots ,N_{c}$ of the database, i.e.,
$$S_{u}^{i}=\left[\begin{array}{c} \left\{u\right\}\left(\left\{X\right\},t^{i},\theta_{1}\right),\;\ldots ,\;\left\{u\right\}\left(\left\{X\right\},t^{i},\theta_{N_{c}}\right) \end{array}\right],$$
and
$$S_{v}^{i}=\left[\begin{array}{c} \left\{v\right\}\left(\left\{X\right\},t^{i},\theta_{1}\right),\;\ldots ,\;\left\{v\right\}\left(\left\{X\right\},t^{i},\theta_{N_{c}}\right) \end{array}\right].$$Then we stack all the matrices $S_{\varrho }^{i}$ for $i=1,\ldots ,N_{t}$ into a big matrix $S_{\varrho }\in M_{3N_{x}\times N_{t},N_{c}}\left(R\right)$:$$S_{\varrho }=\left[\begin{array}{c} S_{\varrho }^{1}\\ S_{\varrho }^{2}\\ \vdots \\ S_{\varrho }^{N_{t}} \end{array}\right]\;\mathrm{for}\;\varrho =u,v.$$We then apply SVD [42] and get:(15)$$S_{\varrho }=U_{\varrho }\;\Sigma_{S_{\varrho }}\;\Psi_{\varrho }^{T},\;\mathrm{for}\;\varrho =u,v,$$
where $U_{\varrho }\in M_{3N_{x}N_{t},N_{c}}\left(R\right)$, $\Psi_{\varrho }\in M_{N_{c}}\left(R\right)$ are semi-orthogonal and orthogonal matrices, respectively, and $\Sigma_{S_{\varrho }}\in M_{N_{c}}\left(R\right)$ is the diagonal singular value matrix. The matrix $\Psi_{\varrho }$ of discrete parameter modes can be truncated according to $K_{\varrho }^{c}$ parameters, so we note:(16)$$\left\{\begin{array}{c} \Psi_{\varrho }^{r}=\left[\begin{array}{c} {\left(\Psi_{\varrho }\right)}_{1},\ldots ,{\left(\Psi_{\varrho }\right)}_{K_{\varrho }^{c}} \end{array}\right]\in M_{N_{c},K_{\varrho }^{c}}\left(R\right),\\ \;\mathrm{with}\;{\left(\Psi_{\varrho }\right)}_{k}\in M_{N_{c},1}\left(R\right)\;\mathrm{for}\;k=1,\ldots ,K_{\varrho }^{c},\;\mathrm{and}\;\varrho =u,v. \end{array}\right.$$The orthogonality property ensures that ${\left(\Psi_{\varrho }^{r}\right)}^{T}\Psi_{\varrho }^{r}=I_{K_{\varrho }^{c}}$.

#### 3.2.2. Global Spatial Reduced Basis (GSRB)

Similarly, we build a global spatial reduced basis that captures the spatial data of capsule displacements. Let $T_{u}^{j}\in M_{3N_{x},N_{t}}\left(R\right)$ be the displacement matrix and $T_{v}^{j}$ the velocity matrix for the *j*-th configuration $\theta_{j}$, for $j=1,\ldots ,N_{c}$ at all time instants $t^{i}$, $i=1,\ldots ,N_{t}$ (Figure 4b):$$T_{u}^{j}=\left[\begin{array}{c} \left\{u\right\}\left(\left\{X\right\},t^{1},\theta_{j}\right),\;\ldots ,\;\left\{u\right\}\left(\left\{X\right\},t^{N_{t}},\theta_{j}\right) \end{array}\right],$$
and
$$T_{v}^{j}=\left[\begin{array}{c} \left\{v\right\}\left(\left\{X\right\},t^{1},\theta_{j}\right),\;\ldots ,\;\left\{v\right\}\left(\left\{X\right\},t^{N_{t}},\theta_{j}\right) \end{array}\right].$$Then we define the global matrix $T_{\varrho }\in M_{3N_{x},N_{t}\times N_{c}}\left(R\right)$ that horizontally gathers all the matrices $T_{\varrho }^{j}$ for $j=1,\ldots ,N_{c}$ and ϱ=u,v, respectively:$$T_{\varrho }=\left[\begin{array}{c} T_{\varrho }^{1},T_{\varrho }^{2},\ldots ,T_{\varrho }^{N_{c}} \end{array}\right],\;\mathrm{for}\;\varrho =u,v.$$The SVD decomposition is applied on $T_{\varrho }$ to get
(17)$$T_{\varrho }=\Phi_{\varrho }\;\Sigma_{T_{\varrho }}\;V_{\varrho }^{T},\;\mathrm{for}\;\varrho =u,v,$$
where $\Phi_{\varrho }\in M_{3N_{x}}\left(R\right)$, $V_{\varrho }\in M_{N_{c}N_{t},3N_{x}}\left(R\right)$ are orthogonal and semi-orthogonal matrices, respectively, and $\Sigma_{S_{\varrho }}\in M_{3N_{x}}\left(R\right)$ is the diagonal singular value matrix with singular values organized in decreasing order. We can also apply a spatial basis truncation at a range of $K_{\varrho }^{x}$ for a specified accuracy threshold. The reduced spatial POD basis is stored in the matrix:(18)$$\Phi_{\varrho }^{r}=\left[\begin{array}{c} {\left\{\phi_{\varrho }\right\}}_{1},\ldots ,{\left\{\phi_{\varrho }\right\}}_{K_{\varrho }^{x}} \end{array}\right]\in M_{3N_{x},K_{\varrho }^{x}}\left(R\right)$$
with the orthogonality property ${\left(\Phi_{\varrho }^{r}\right)}^{T}\Phi_{\varrho }^{r}=I_{K_{\varrho }^{x}}$, ϱ=u,v.

### 3.3. Data Dimensionality Reduction

Once the POD modes of $S_{\varrho }$ and $T_{\varrho }$ for the displacement fields $\left(\varrho =u\right)$ and the velocity fields $\left(\varrho =v\right)$ are computed, one can summarize (approximate) capsule displacement and velocity fields of the database at any discrete time $t^{i}$ ($i=1\ldots ,N_{t}$) as
(19)$$\left\{u\right\}\left(\left\{X\right\},\;t^{i},\;\left[\theta_{1},\ldots ,\theta_{N_{c}}\right]\right)\approx \Phi_{u}^{r}\;A\left(t^{i}\right)\;{\left(\Psi_{u}^{r}\right)}^{T}\in M_{3N_{x},N_{c}}\left(R\right),$$
(20)$$\left\{v\right\}\left(\left\{X\right\},t^{i},\;\left[\theta_{1},\ldots ,\theta_{N_{c}}\right]\right)\approx \Phi_{v}^{r}\;B\left(t^{i}\right)\;{\left(\Psi_{v}^{r}\right)}^{T}\in M_{3N_{x},N_{c}}\left(R\right),$$
where $A\left(t^{i}\right)\in M_{K_{u}^{x},K_{u}^{c}}\left(R\right)$ and $B\left(t^{i}\right)\in M_{K_{v}^{x},K_{v}^{c}}\left(R\right)$ are some coefficient matrices depending on time $t^{i}$. If the approximation is chosen as the orthogonal projection over the vector spaces spanned by the POD modes, the coefficient matrices are computed as follows for $i=1\ldots ,N_{t}$:(21)$$A\left(t^{i}\right)=\underset{K_{u}^{x}\times \left(3N_{x}\right)}{\underset{︸}{{\left(\Phi_{u}^{r}\right)}^{T}}}\;\underset{\left(3N_{x}\right)\times N_{c}}{\underset{︸}{\left\{u\right\}\left(\left\{X\right\},t^{i},\;\left[\theta_{1},\ldots ,\theta_{N_{c}}\right]\right)}}\;\underset{N_{c}\times K_{u}^{c}}{\underset{︸}{\Psi_{u}^{r}}},$$
(22)$$B\left(t^{i}\right)=\underset{K_{v}^{x}\times \left(3N_{x}\right)}{\underset{︸}{{\left(\Phi_{v}^{r}\right)}^{T}}}\;\underset{\left(3N_{x}\right)\times N_{c}}{\underset{︸}{\left\{v\right\}\left(\left\{X\right\},t^{i},\;\left[\theta_{1},\ldots ,\theta_{N_{c}}\right]\right)}}\;\underset{N_{c}\times K_{v}^{c}}{\underset{︸}{\Psi_{v}^{r}}}.$$

The outputs of the offline stage are respectively the POD matrices $\Phi_{u}^{r}$, $\Phi_{v}^{r}$, $\Psi_{u}^{r}$, $\Psi_{v}^{r}$ and the small matrices $A(t^{i})$, $B(t^{i})$, $i=1,\ldots ,N_{t}$. The next online stage will operate on the summarized data (Equation 19), (Equation 20) with coefficients matrices (Equation 21), (Equation 22). The algorithm of the offline phase is summarized in Algorithm 1.
**Algorithm 1** Offline phase**Require:** database of $\theta_{k}$ for $k=1,\ldots ,N_{c}$, truncations $K_{\varrho }^{c}$, number of snapshots $N_{t}$. **for** $i\leftarrow 1,\ldots ,N_{t}$ **do**  **if** (ϱ=u) **then**   $S_{u}^{i}\leftarrow \left[\begin{array}{c} \left\{u\right\}\left(\left\{X\right\},t^{i},\theta_{1}\right),\;\ldots ,\;\left\{u\right\}\left(\left\{X\right\},t^{i},\theta_{N_{c}}\right) \end{array}\right]$; $S_{u}\leftarrow \left[S_{u};S_{u}^{i}\right]$;  **else**   $S_{v}^{i}\leftarrow \left[\begin{array}{c} \left\{v\right\}\left(\left\{X\right\},t^{i},\theta_{1}\right),\;\ldots ,\;\left\{v\right\}\left(\left\{X\right\},t^{i},\theta_{N_{c}}\right) \end{array}\right]$; $S_{v}\leftarrow \left[S_{v},S_{v}^{i}\right]$;  **end if** **end for** **for** $j\leftarrow 1,\ldots ,N_{c}$ **do**  **if** (ϱ=u) **then**   $T_{u}^{j}\leftarrow \left[\begin{array}{c} \left\{u\right\}\left(\left\{X\right\},t^{1},\theta_{j}\right),\;\ldots ,\;\left\{u\right\}\left(\left\{X\right\},t^{N_{t}},\theta_{j}\right) \end{array}\right]$; $T_{u}\leftarrow \left[T_{u},T_{u}^{j}\right]$;  **else**   $T_{v}^{j}\leftarrow \left[\begin{array}{c} \left\{v\right\}\left(\left\{X\right\},t^{1},\theta_{j}\right),\;\ldots ,\;\left\{v\right\}\left(\left\{x\right\},t^{N_{t}},\theta_{j}\right) \end{array}\right]$; $T_{v}\leftarrow \left[T_{v},T_{v}^{j}\right]$;  **end if** **end for** $\Phi_{\varrho }\leftarrow$ SVD($S_{\varrho }$), $\Psi_{\varrho }\leftarrow$ SVD($T_{\varrho }$), for ϱ←u,v; **for** $i=1,\ldots ,N_{t}$ **do**  $A\left(t^{i}\right)\leftarrow {\left(\Phi_{u}^{r}\right)}^{T}\;\left\{u\right\}\left(\left\{X\right\},t^{i},\;\left[\theta_{1},\ldots ,\theta_{N_{c}}\right]\right)\;\Psi_{u}^{r}$;  $B\left(t^{i}\right)\leftarrow {\left(\Phi_{v}^{r}\right)}^{T}\;\left\{v\right\}\left(\left\{X\right\},t^{i},\left[\theta_{1},\ldots ,\theta_{N_{c}}\right]\right)\;\Psi_{v}^{r}$; **end for**

### 3.4. Online Stage: Search for an Approximate Solution at a
Query Configuration $\theta_{q}$

In the online stage, a user will ask for an approximate solution at a new (query) configuration $\theta =\theta_{q}$ that has not been already computed by the FOM solver or is not stored in the database. Ingredients of the online stage will be: (i) the data summarization of the previous offline stage; (ii) a first estimation of the spatio-temporal solution at $\theta =\theta_{q}$; (iii) the computation of a low-dimensional spatial reduced basis suitable for $\theta =\theta_{q}$; (iv) the construction of a manifold M that links variables of displacements and velocities in the low-order state space to solve the equation of membrane mechanics; (v) finally, the building of a low-order differential-algebraic (DAE) system of equations that defines the reduced-order model. Substeps (ii) and (iv) will make use of diffuse approximation (DA) as a universal approximator for multivariate functions.

#### 3.4.1. First Estimation of the Solutions at $\theta =\theta_{q}$

As an introduction, let us assume that, from the parameter sampling $\left\{\theta_{1},\ldots ,\theta_{N_{c}}\right\}$, we consider a polynomial Lagrange interpolation with Lagrange polynomials denoted by $L_{i}\left(\theta \right)$ such that the Lagrange property
$$L_{i}\left(\theta_{j}\right)=\delta_{ij},\;1\leq i,j\leq N_{c}$$
is fulfilled ($\delta_{ij}$ is the standard Kronecker symbol). Let us denote by $L\left(\theta \right)={\left(L_{j}\left(\theta \right)\right)}_{j=1,\ldots ,N_{c}}\in R^{N_{c}}$ the vector that stores the Lagrange polynomials. Then
$$I\left\{u\right\}\left(\left\{X\right\},\;t^{i},\;\theta_{q}\right):=\left\{u\right\}\left(\left\{X\right\},\;t^{i},\;\left[\theta_{1},\ldots ,\theta_{N_{c}}\right]\right)L\left(\theta_{q}\right)\in R^{3N_{x}}$$
is an interpolated displacement field at parameter $\theta =\theta_{q}$ and discrete time $t=t^{i}$. One can of course do the same for the velocity field.

Unfortunately, Lagrange polynomial interpolation is not suitable for parameter spaces of arbitrary dimension because of the curse of dimensionality and because it may suffer from instability issues (Runge phenomenon). Rather than using polynomial interpolation, we propose to use a Diffuse Approximation (DA) technique [18,29] which is an approximation method based on local low-order polynomial reconstruction (of order one or two) using a compactly-supported kernel function and weighted least squares. The DA method is known to be a robust and reliable approach which is less sensitive to the location of the sampling points. Moreover, it can be applied to multivariate functions of arbitrary dimensions, which is interesting for larger or more general parameter spaces. It is particularly suited for the current problem, for which the sampling is performed on a Cartesian grid. It may fail in the occurrence of local point alignment within the cloud points, which does not occur in the present study. The accuracy of the DA method may slightly decrease close to the boundary of the domain, the number of neighboring points being reduced.

To estimate the displacement field for $\theta =\theta_{q}$, we look for a vector $\psi_{u}\left(\theta_{q}\right)\in R^{K_{u}^{c}}$ such that
(23)$$\left\{u\right\}\left(\left\{x\right\},t^{i},\theta_{q}\right)=\Phi_{u}^{r}\;A\left(t^{i}\right)\;\psi_{u}\left(\theta_{q}\right)$$
returns an approximation of the displacement field at $\theta =\theta_{q}$. Similarly for the velocity field, we search for a vector $\psi_{v}\left(\theta_{q}\right)\in R^{K_{v}^{c}}$ that gives
(24)$$\left\{v\right\}\left(\left\{x\right\},t^{i},\theta_{q}\right)=\Phi_{v}^{r}\;B\left(t^{i}\right)\;\psi_{v}\left(\theta_{q}\right).$$Each vector $\psi_{\varrho }\left(\theta_{q}\right)\in R^{K_{\varrho }^{c}}$ can be locally approximated by
(25)$$\Psi_{\varrho }\left(\theta_{q}\right)=A_{\varrho }\;p\left(\theta_{q}\right),\;\mathrm{for}\;\varrho =u,v,$$
where the matrix $A_{\varrho }\in M_{K_{\varrho }^{c},m}\left(R\right)$ (to be determined) is the approximation coefficient matrix and $p\left(\theta_{q}\right)\;\in R^{m}$ is a vector of independent polynomial functions, where
(26)$$\left\{\begin{array}{cc} p\left(\theta \right)\;={\left[\begin{array}{ccc} 1 & Ca & a/\ell \end{array}\right]}^{T},\;m=3 & \;\mathrm{for}\;\mathrm{first}\;\mathrm{order}\;\mathrm{DA},\\ p\left(\theta \right)\;={\left[\begin{array}{cccccc} 1 & Ca & a/\ell & Ca\left(a/\ell \right) & {\left(Ca\right)}^{2} & {\left(a/\ell \right)}^{2} \end{array}\right]}^{T},\;m=6 & \;\mathrm{for}\;\mathrm{second}\;\mathrm{order}\;\mathrm{DA}. \end{array}\right.$$

To approximate $\psi_{\varrho }\left(\theta_{q}\right)$, let us consider a neighborhood $S(\theta_{q})$ centered on $\theta_{q}$ containing *M* neighboring points (Figure 5a). It is an ellipse of equation
$${\left(\theta_{1}-{\left(\theta_{q}\right)}_{1}\right)}^{2}+{\tilde{r}}^{2}{\left(\theta_{2}-{\left(\theta_{q}\right)}_{2}\right)}^{2}=R^{2}$$
where $\tilde{r}$ is fixed (equal to 1.9 in Figure 5a) and *R* is chosen such that the ellipse contains *M* points (*M* being chosen by the operator). In other words, the distance between $\theta ={\left(\theta_{1},\theta_{2}\right)}^{T}$ and $\theta_{q}$ is
(27)$$d={\left({\left(\theta_{1}-{\left(\theta_{q}\right)}_{1}\right)}^{2}+{\tilde{r}}^{2}{\left(\theta_{2}-{\left(\theta_{q}\right)}_{2}\right)}^{2}\right)}^{\frac{1}{2}}/R.$$The compactly supported Wendland weight function shown in Figure 5b is classically used. It has appropriate high-order approximation properties ([43]):(28)$$\left\{\begin{array}{cc} w\left(d\right)=2\;d^{3}-3\;d^{2}+1, & \;d\leq 1,\\ 0, & \;\mathrm{otherwise}. \end{array}\right.$$Diffuse approximation consists in minimizing the weighted least square problem
(29)$$\min_{A_{\varrho }\in M_{K_{\varrho }^{c},m}\left(R\right)}\;J_{\theta_{q}}\left(A_{\varrho }\right):=\frac{1}{2}\sum_{\theta \in S(\theta_{q})}w\left(d(\theta )\right){∥A_{\varrho }\;p\left(\theta \right)-{\left[\Psi_{\varrho }^{r}\left(\theta \right)\right]}^{T}∥}_{R^{K_{\varrho }^{c}}}^{2}$$
where ${\left[\Psi_{\varrho }^{r}\left(\theta \right)\right]}^{T}$ is the truncated matrix of modes that correspond to couples $\theta_{k}$, $k=1,\ldots ,N_{c}$. The solution $A_{\varrho }$(ϱ=u,v) of the weighted least square problem (Equation 29) is then
(30)$$A_{\varrho }={\left(\Psi_{\varrho }^{r}\right)}^{T}WP{\left(P^{T}WP\right)}^{-1}\in M_{K_{\varrho }^{c},m}\left(R\right)$$
where the matrix $P\in M_{N_{c},m}\left(R\right)$ and the diagonal matrix of weights $W\in M_{N_{c}}\left(R\right)$ are defined as
(31)$$P=\left[\begin{array}{c} p{\left(\theta_{1}\right)}^{T}\\ \vdots \\ p{\left(\theta_{N_{c}}\right)}^{T} \end{array}\right]\;\mathrm{and}\;W=\left[\begin{array}{cccc} w_{1} & 0 & \cdots & 0\\ 0 & w_{2} & \; & \vdots \\ \vdots & \; & \ddots & \\ 0 & \cdots & \; & w_{N_{c}} \end{array}\right].$$

#### 3.4.2. Construction of a Low-Order Reduced Basis Suitable for $\theta =\theta_{q}$, Data Generation

From (Equation 23) and (Equation 24), one can easily generate some pseudo-snapshot matrices $U(\theta_{q})$ and $V(\theta_{q})$ that gather the estimated displacements and velocities at $N_{t}$ discrete times, respectively:(32)$$\left\{\begin{array}{c} U\left(\theta_{q}\right)=\left[\begin{array}{c} \left\{u\right\}\left(\left\{X\right\},t^{1},\theta_{q}\right),\ldots ,\left\{u\right\}\left(\left\{X\right\},t^{N_{t}},\theta_{q}\right) \end{array}\right],\\ V\left(\theta_{q}\right)=\left[\begin{array}{c} \left\{v\right\}\left(\left\{X\right\},t^{1},\theta_{q}\right),\ldots ,\left\{v\right\}\left(\left\{X\right\},t^{N_{t}},\theta_{q}\right) \end{array}\right]. \end{array}\right.$$

One can then apply a new SVD decomposition of matrices $U(\theta_{q})$ and $V(\theta_{q})$ respectively to get spatial POD modes $\left\{\varphi^{k}\right\}\left(\theta_{q}\right)\in R^{3N_{x}}$, $k=1,\ldots ,m_{u}$ for {u} and velocity POD modes $\left\{\gamma^{k}\right\}\left(\theta_{q}\right)\in R^{3N_{x}}$, $k=1,\ldots ,m_{v}$ for {v}.
(33)$$\begin{array}{ccc} \mathrm{POD}\left(U(\theta_{q})\right) & \to & \left\{\varphi^{1}\right\}\left(\theta_{q}\right),\ldots ,\left\{\varphi^{m_{u}}\right\}\left(\theta_{q}\right) \end{array}$$
(34)$$\begin{array}{ccc} \mathrm{POD}\left(V(\theta_{q})\right) & \to & \left\{\gamma^{1}\right\}\left(\theta_{q}\right),\ldots ,\left\{\gamma^{m_{v}}\right\}\left(\theta_{q}\right) \end{array}$$
where $m_{u}$ and $m_{v}$ are the truncation ranks of displacement and velocity modes determined in the next section on numerical experiments. One can then search the displacement and velocity fields at $\theta =\theta_{q}$ as
(35)$$\left\{u\right\}\left(\left\{X\right\},t,\theta_{q}\right)=\underset{k=1}{\sum^{m_{u}}}\alpha_{k}\left(t\right)\left\{\varphi^{k}\right\}\left(\theta_{q}\right),$$
(36)$$\left\{v\right\}\left(\left\{X\right\},t,\theta_{q}\right)=\underset{k=1}{\sum^{m_{v}}}\xi_{k}\left(t\right)\left\{\gamma^{k}\right\}\left(\theta_{q}\right).$$By denoting
(37)$$\begin{array}{cc} \; & \Phi \left(\theta_{q}\right)=\left[\left\{\varphi^{1}\right\}\left(\theta_{q}\right),\ldots ,\left\{\varphi^{m_{u}}\right\}\left(\theta_{q}\right)\right]\in M_{3Nx,m_{u}}\left(R\right), \end{array}$$
(38)$$\begin{array}{cc} \; & \Gamma \left(\theta_{q}\right)=\left[\left\{\gamma^{1}\right\}\left(\theta_{q}\right),\ldots ,\left\{\gamma^{m_{v}}\right\}\left(\theta_{q}\right)\right]\in M_{3Nx,m_{v}}\left(R\right) \end{array}$$
and $\alpha \left(t\right)={\left[\alpha_{1}\left(t\right),\ldots ,\alpha_{m_{u}}\left(t\right)\right]}^{T}\in R^{m_{u}}$, $\xi \left(t\right)={\left[\xi_{1}\left(t\right),\ldots ,\xi_{m_{v}}\left(t\right)\right]}^{T}\in R^{m_{v}}$, we have the vector formulas
(39)$$\left\{u\right\}\left(\left\{X\right\},t,\theta_{q}\right)=\Phi \left(\theta_{q}\right)\;\alpha \left(t\right),\;\left\{v\right\}\left(\left\{X\right\},t,\theta_{q}\right)=\Gamma \left(\theta_{q}\right)\;\xi \left(t\right).$$

The mode matrices $\Phi (\theta_{q})$ and $\Gamma (\theta_{q})$ are assumed to be orthonormal (w.r.t the natural Euclidean inner product), so we have ${\left[\Phi \left(\theta_{q}\right)\right]}^{T}\;\Phi \left(\theta_{q}\right)=I_{m_{u}}$ and ${\left[\Gamma \left(\theta_{q}\right)\right]}^{T}\;\Gamma \left(\theta_{q}\right)=I_{m_{v}}$.

#### 3.4.3. Toward a Physically Consistent Dynamical Reduced-Order Model

Consider now the forward Euler scheme on the FSI system with a ROM time step $\delta t^{ROM}>0$: at time $t^{i+1,ROM}=t^{i,ROM}+\delta t^{ROM}$, the numerical scheme is
(40)$$\begin{array}{cc} \; & \left\{u^{i+1}\right\}=\left\{u^{i}\right\}+\delta t^{ROM}\;\left\{v^{i}\right\}, \end{array}$$
(41)$$\begin{array}{cc} \; & \left\{v^{i+1}\right\}=\left\{N\right\}\left(\left\{u^{i+1}\right\},\;\theta_{q}\right). \end{array}$$Let us emphasize that the equation of local mechanical equilibrium depends on the parameter $\theta_{q}$. For the reduced-order model, we would like to have a similar algebraic structure to (Equation 40), (Equation 41) but formulated as a low-dimensional system. If $\left\{u^{i}\right\}$ and $\left\{v^{i}\right\}$ are searched in the form $\left\{u^{i}\right\}=\Phi \left(\theta_{q}\right)\;\alpha^{i}$ and $\left\{v^{i}\right\}=\Gamma \left(\theta_{q}\right)\;\xi^{i}$, respectively, Equation (Equation 40) becomes
$$\Phi \left(\theta_{q}\right)\;\alpha^{i+1}=\Phi \left(\theta_{q}\right)\;\alpha^{i}+\delta t^{ROM}\;\Gamma \left(\theta_{q}\right)\;\xi^{i}.$$By multiplying by ${\left[\Phi \left(\theta_{q}\right)\right]}^{T}$ on the left, we get the system of $m_{u}$ equations
(42)$$\alpha^{i+1}=\alpha^{i}+\delta t^{ROM}\;Q\left(\theta_{q}\right)\;\xi^{i},$$
where $Q\left(\theta_{q}\right)={\left[\Phi \left(\theta_{q}\right)\right]}^{T}\Gamma \left(\theta_{q}\right)$. Equation (Equation 41) is replaced by
$$\Gamma \left(\theta_{q}\right)\;\xi^{i+1}=\left\{N\right\}\left(\Phi \left(\theta_{q}\right)\;\alpha^{i+1},\theta_{q}\right).$$By multiplying by ${\left[\Gamma \left(\theta_{q}\right)\right]}^{T}$ on the left, we get
$$\xi^{i+1}=M\left(\alpha^{i+1},\theta_{q}\right)$$
where
(43)$$M\left(\alpha^{i+1},\theta_{q}\right)={\left[\Gamma \left(\theta_{q}\right)\right]}^{T}\left\{N\right\}\left(\Phi \left(\theta_{q}\right)\;\alpha^{i+1},\theta_{q}\right)\in R^{m_{v}}.$$

#### 3.4.4. Manifold Learning

Because of nonlinear terms, the direct computation of $M(\alpha^{i+1},\theta_{q})$ in (Equation 43) requires high-dimensional computations, which makes the ROM irrelevant from a performance point of view. To “identify” a low-order manifold M, we rather adopt a data-driven approach based once again of diffuse approximation. We link the entry data $\alpha_{k}^{D}\left(t^{i}\right)$, $k=1,\ldots ,m_{u}$, $i=1,\ldots ,N_{t}$ to the output data $\xi_{k}^{D}\left(t^{i}\right)$, $k=1,\ldots ,m_{v}$, $i=1,\ldots ,N_{t}$ (*’D’* stands for *’data’*). For that, one can compute the orthogonal projections of the pseudo-snapshots over the POD bases, leading to the formulas
$$\alpha_{k}^{D}\left(t^{i}\right)=\langle \left\{u\right\}\left(\left\{X\right\},t^{i},\theta_{q}\right),\;\left\{\varphi^{k}\right\}\left(\theta_{q}\right)\rangle$$
and
$$\xi^{D}\left(t^{i}\right)=\langle \left\{v\right\}\left(\left\{X\right\},t^{i},\theta_{q}\right),\;\left\{\gamma^{k}\right\}\left(\theta_{q}\right)\rangle$$
at instants $t^{i}=i\Delta t$. Manifold learning consists in achieving a (nonlinear) regression method that links entry and output data. We are looking for a manifold representation $\xi =M(\alpha ,\theta_{q})$ in the form
(44)$$\xi_{k}=p{\left(\alpha \right)}^{T}a^{k},\;k=1,\ldots ,m_{v}$$
where p(α) is the vector made of monomials in α of order zero and one, and $a^{k}\in R^{m_{u}+1}$ is a vector of coefficients to be determined from the data. This corresponds to a local linear embedding process. For each $k=1,\ldots ,m_{v}$, one looks for a coefficient vector $a^{k}\left(t\right)\in R^{m_{u}+1}$ solution of the weighted least square problem
(45)$$a^{k}\left(t\right)=\arg\min_{a\in R^{m_{u}+1}}\;\frac{1}{2}\sum_{i=1}^{N_{t}}w\left(\frac{|t-t^{i}|}{R}\right)\;{\left(p{\left(\alpha^{D}\left(t^{i}\right)\right)}^{T}\;a-\xi_{k}^{D}\left(t^{i}\right)\right)}^{2}$$
where $t\in [0,T_{f}]$, w=w(d) is the weight function defined in Figure 5b and $d=\frac{|t-t^{i}|}{R}$. This returns a regression function
(46)$$\xi_{k}=\xi_{k}\left(t,\alpha \left(t\right)\right)=p{\left(\alpha \left(t\right)\right)}^{T}\;a^{k}\left(t\right).$$

#### 3.4.5. Low-Order Dynamical Reduced Order Model

The resulting time-discrete reduced-order model is then
(47)$$\begin{array}{cc} \; & t^{i+1,ROM}=t^{i,ROM}+\delta t^{ROM}, \end{array}$$
(48)$$\begin{array}{cc} \; & \alpha^{i+1}=\alpha^{i}+\delta t^{ROM}\;Q\left(\theta_{q}\right)\;\xi^{i}, \end{array}$$
(49)$$\begin{array}{cc} \; & \xi_{k}^{i+1}=p{\left(\alpha^{i+1}\right)}^{T}\;a^{k}\left(t^{i+1,ROM}\right)\;\forall k\in \left\{1,\ldots ,m_{v}\right\}. \end{array}$$High-dimensional displacement and velocity fields can be reconstructed as follows:$$\left\{u\right\}\left(\left\{X\right\},t^{i+1,ROM},\theta_{q}\right)=\Phi \left(\theta_{q}\right)\;\alpha^{i+1},\;\left\{v\right\}\left(\left\{X\right\},t^{i+1,ROM},\theta_{q}\right)=\Gamma \left(\theta_{q}\right)\;\xi^{i+1}.$$The online stage of the reduced-order model is summarized in Algorithm 2.
**Algorithm 2** Online phase**Require:** choose a query parameter $\theta_{q}$, choose a time step $\delta t^{ROM}>0$. Initialization: $t=t^{0,ROM}=0$, $\alpha^{0}=0$, $\xi^{0}=\xi^{D}\left(0\right)$; Compute $\Psi_{u}\left(\theta_{q}\right)$ and $\Psi_{v}\left(\theta_{q}\right)$ from the diffuse approximation approach; **for** $i=1\ldots ,N_{t}$ **do**  $\left\{u\right\}\left(\left\{x\right\},t^{i},\theta_{q}\right)\leftarrow \Phi_{u}^{r}\;A\left(t^{i}\right)\;\Psi_{u}\left(\theta_{q}\right)$;  $\left\{v\right\}\left(\left\{x\right\},t^{i},\theta_{q}\right)\leftarrow \Phi_{v}^{r}\;B\left(t^{i}\right)\;\Psi_{v}\left(\theta_{q}\right)$; **end for** $U\left(\theta_{q}\right)\leftarrow \left[\left\{u\right\}\left(\left\{x\right\},t^{1},\theta_{q}\right),\ldots ,\left\{u\right\}\left(\left\{x\right\},t^{N_{t}},\theta_{q}\right)\right]$; $V\left(\theta_{q}\right)\leftarrow \left[\left\{v\right\}\left(\left\{x\right\},t^{1},\theta_{q}\right),\ldots ,\left\{v\right\}\left(\left\{x\right\},t^{N_{t}},\theta_{q}\right)\right]$; Compute $\Phi (\theta_{q})$, $\Gamma (\theta_{q})$, $Q(\theta_{q})$, $\alpha^{D}\left(t^{i}\right)$ and $\xi^{D}\left(\theta_{i}\right)$, $i=1,\ldots ,N_{t}$; **while** $t<T_{f}$ **do**  $t\leftarrow t+\delta t^{ROM}$; $t^{i+1,ROM}=t^{i,ROM}+\delta t^{ROM}$;  $\alpha^{i+1}=\alpha^{i}+\delta t^{ROM}\;Q\left(\theta_{q}\right)\xi^{i}$;  Compute $a^{k}\left(t^{i+1,ROM}\right)$, $k=1,\ldots ,m_{v}$ from the diffuse approximation approach;  $\xi_{k}^{i+1}=p{\left(\alpha^{i+1}\right)}^{T}\;a^{k}\left(t^{i+1,ROM}\right)$;  If needed, reconstruct the high-dimensional displacements/velocity fields:  $\left\{u\right\}\left(\left\{X\right\},t^{i+1,ROM},\theta_{q}\right)=\Phi \left(\theta_{q}\right)\;\alpha^{i+1}$;  $\left\{v\right\}\left(\left\{X\right\},t^{i+1,ROM},\theta_{q}\right)=\Gamma \left(\theta_{q}\right)\;\xi^{i+1}$;**end while**

## 4. Numerical Experiments

### 4.1. Study Case

We consider a capsule flowing in a square-base microchannel of base edges of length 2ℓ. We want to capture the capsule dynamics for capillary numbers Ca belonging to the interval [0.005,0.2] and aspect ratios a/ℓ in the interval [0.75,1.2] for which a steady state shape is reached. The Caps3D code [38,40] is then used as FOM solver. The comparison of the FOM results with experimental ones using a square-base cylinder have been thoroughly described in other previous studies (see for example [23,24,27,39] from A.V. Salsac’s research team). The total non-dimensional time for simulation is T=20. For any capillary number and aspect ratio, the capsule is discretized with the same mesh resolution and connectivity, consisting of $N_{x}=2562$ nodes (corresponding to 1280 triangular elements), with a capsule mesh size $\Delta h_{C}=0.075\;a$ (see Figure 6). A second-order RK2 Ralston scheme is used for time integration. The dimensionless time step is $\dot{\gamma }\delta t=5\cdot 10^{-4}$ for Ca>0.02 and $\dot{\gamma }\delta t=10^{-4}$ for Ca≤0.02 .

### 4.2. FOM Result Database Generation

A database of FOM results is generated from a sampling of the parameter domain (see Figure 7). It is observed that configurations for which a shape steady state is reached before the non-dimensional final time of 20 correspond to couples (Ca,a/ℓ) in the parameter plane below the dashed red line of Figure 7. Using a Cartesian parameter sampling with step sizes of 0.01 in Ca and 0.05 in a/ℓ, plus few additional points at Ca=0.005, we get a database made of $N_{c}=118$ configurations. From Caps3D FOM solutions, we pick up time-snapshot solutions every time step Δt=0.2 in non-dimensional time scale, corresponding to $N_{t}=100$. This makes a datacube made of $2\times 3N_{x}N_{c}N_{t}\approx 1.81\cdot 10^{8}$ double precision float numbers taking about 1.45 GB of memory.

#### Clustering Strategy

For the sake of memory storage complexity, we adopt a strategy of data clustering with two weakly-overlapping clusters chosen manually, represented in Figure 7. For each cluster, a data dimensionality reduction is done following the offline-stage algorithm presented in Section 3. That means that two families of reduced-order models are actually computed. In the online stage, for a new query parameter vector $\theta_{q}$, one has to determine the cluster of belonging.

### 4.3. Elements of Analysis—Accuracy Criteria

In order to measure the approximation error generated by the data dimensionality process, we introduce the classical Relative Information Content (RIC) (see for example [9]), which is computed as:(50)$$\mathrm{RIC}\left(K\right)=\frac{\sum_{k=K+1}^{r}{\tilde{\sigma }}_{k}^{2}}{\sum_{k=1}^{r}{\tilde{\sigma }}_{k}^{2}},\;K=1,\ldots ,r,$$
where $\tilde{\sigma_{k}}$ is the *k*-th singular value from the SVD decomposition, *r* is the rank of the matrix of study ($S_{\varrho }$ or $T_{\varrho }$) and *K* is the truncation rank. A supplementary indicator is the ratio
(51)$$K\mapsto \frac{{\tilde{\sigma }}_{K}}{{\tilde{\sigma }}_{1}}$$
that gives an idea of the decay rate of the singular values.

The second criterion directly measures the error between the shape predicted by the ROM and the shape computed by the FOM. This is achieved by using the so-called Modified Hausdorff distance $d_{MH}$ [44] that we normalize by the capsule radius *a*. The modified Hausdorff distance computes the distance between two finite sets F and G of a normed space of norm ∥.∥, and is defined as
(52)$$d_{MH}\left(F,G\right)=\max\left(d_{h}\left(F,G\right),d_{h}\left(F,G\right)\right),$$
with
(53)$$d_{h}\left(F,G\right)=\frac{1}{N_{F}}\sum_{p_{F}\in F}d_{s}\left(p_{F},G\right)$$
where $N_{F}$ is the number of points in the set F and $d_{s}\left(p_{F},G\right)$ is the distance between $p_{F}$ and the set G, which is defined as
(54)$$d_{s}\left(p_{F},G\right)=\min_{p_{G}\in G}\;∥p_{F}-p_{G}∥.$$

### 4.4. Dimensionality Reduction Analysis

A singular value decomposition analysis is first performed on the matrices $S_{u}$ and $S_{v}$, and then on $T_{u}$ and $T_{v}$. In Figure 8a, we plot the indicator (1−RIC) (see (Equation 50)), as a function of the truncation rank *K*, for $S_{u}$ and $S_{v}$. What can be seen is that (1−RIC) rapidly converges towards the value 0 in all cases. An expected (1−RIC) of $10^{-7}$ is reached for a truncation rank $K_{u}^{c}$ (resp. $K_{v}^{c}$) of 7 for the displacement (resp. 23 for the velocity). Similarly in Figure 8b, we plot the indicator (1−RIC) for $T_{u}$ and $T_{v}$. The number of modes $K_{u}^{x}$ (resp. $K_{v}^{x}$) needed to reach the threshold of $10^{-7}$ is 7 for the displacement (resp. 56 for the velocity).

As supplementary indicators, the singular values ${\tilde{\sigma }}_{K}$ normalized by ${\tilde{\sigma }}_{1}$ are plotted in Figure 9a (resp. Figure 9b) for both matrices $S_{u}$ and $S_{v}$ (resp. $T_{u}$ and $T_{v}$) in $\log_{10}$ scale. One can first observe a lower decay rate for the velocity fields compared to the displacements, meaning a greater information complexity for the velocity. Secondly, the decay rate is lower for the global spatial mode than for the parametric modes, indicating a larger entropy of information on the whole parameter domain. That justifies the derivation of suitable lower order spatial basis at a query parameter $\theta_{q}$ in the online stage.

At the beginning of the online stage, for a query parameter $\theta_{q}$, an interpolated approximate solution is computed thanks to a diffuse approximation reconstruction. This allows us to get pseudo-snapshots in time for both displacements and velocities, stored in matrices $U(\theta_{q})$ and $V(\theta_{q})$, respectively. We assess the RIC for the two matrices, from an experimental parameter vector $\theta_{q}=\left(0.10,0.90\right)$. The comparison of the time evolution of POD coefficients between FOM and ROM models shows a high accuracy (see Figure 10). and Figure 11 shows that the RIC rapidly converges to 1. An expected RIC greater than $1-10^{-7}$ returns a truncation rank $m_{u}$ (resp. $m_{v}$) of value 3 (resp. 8).

### 4.5. ROM Accuracy Analysis

The reduced-order model algorithm is applied with the following parameters and options:-For global POD modes: $K_{u}^{x}=40$, $K_{u}^{c}=40$, $K_{v}^{x}=50$, $K_{v}^{c}=50$;-For DA in (Equation 25), (Equation 30): local second order polynomial reconstruction, M=12;-For local POD modes: $m_{u}=10$, $m_{v}=10$;-For DA in (Equation 45), (Equation 46): local first order polynomial reconstruction, R=2Δt.

The resulting time-evolution of the three-dimensional capsule shape, that is reconstructed with the ROM model, is illustrated in Figure 12 for the query couple $\theta_{q}=\left(0.10,0.90\right)$. The steady-state is reached before $\dot{\gamma }t=3$, which explains that the capsule shape is the same for $\dot{\gamma }t=3,6,9$.

We now focus on the accuracy analysis of the proposed reduced-order model. The methodology for error measurement is based on a ‘*Leave-one-out*’ cross-validation procedure, where each sample FOM solution is taken out from the database and then evaluated by the ROM model and compared to the original FOM one. The error is measured using the modified Hausdorff distance calculated on the capsule shapes at different instants.

Figure 13 shows the heat maps of the FOM-vs-ROM error computed over the parameter space at the time instants $\dot{\gamma }t=$ 1, 2, 4 and 8. Figure 13 shows that the predicted ROM solutions are very accurate with a mean relative error below 0.2%. The maximum relative errors are below 3.5%: they occur along the boundary of the parameter domain, which is the only location where the predictions slightly lose in accuracy. This is probably due to a lack of well-distributed neighbors close to the boundaries, which affects the accuracy of the DA reconstruction (off-centre approximation). One can also notice that the accuracy of predictions decreases in time.

The capsule cross-section profiles predicted by the ROM (red dots) are compared to the reference FOM solutions (solid black line) in Figure 14 at different time instants ($\dot{\gamma }t=$ 0, 1, 2 and 8) for the 6 configurations, selected as illustration in Figure 3. We observe that the reduced-order model returns very accurate solutions in terms of capsule shape as well-as axial position in the channel.

From the computing performance point of view, ROM-vs-FOM speedups are observed to be of order 10,000 with almost the same accuracy, making interactive exploration and real-time visual rendering possible.

### 4.6. CapsuleExplorer: Capsule Visualization/Exploration Software

We have developed an in-house software tool CapsuleExplorer based on the proposed ROM to provide the three-dimensional microcapsule deformation/evolution at any time $\dot{\gamma }t$ and for any $\theta_{q}$ in the admissible parameter domain. CapsuleExplorer allows one to select a particular couple (Ca,a/ℓ) in the admissible parameter domain, then to visualize the capsule dynamics between initial and final times, either in three dimensions or two dimensions with longitudinal or transversal cross-sectional view. The ROM high performance feature allows real-time exploration/visualization. CapsuleExplorer has been developed as a web application. Figure 15 and Figure 16 show some screenshots of the graphics user interface, which will be useful for applications such as identifying the capsule wall mechanical properties through comparison with experimental results.

## 5. Concluding Remarks

In this paper, we have presented an innovative data-driven reduced-order model that enables the dynamics of a deformable membrane flowing in a microchannel, from its initial state to the steady shape state. The ROM is built to be valid in a large domain of interest in the parameter plane (Ca,a/ℓ). Our FSI-ROM model first starts with an offline procedure to build two global orthonormal bases (space+parameter) that return good approximations of the FOM solutions over the whole parameter domain. The rather small truncation ranks already lead to an appreciable data dimensionality reduction, which is important for complexity and memory storage purposes.

The online stage consists in predicting the space-time solution for any query couple $\theta_{q}=\left(Ca\;,a/\ell \right)$ in the parameter domain. In a first step, we determine a low-order basis for both the displacement and velocity vector variables. This is achieved by the use of diffuse approximation that returns an interpolated space-time solution at the query vector $\theta_{q}$. Then an SVD analysis provides a suitable low-order spatial basis for final construction of the ROM in the second step. The physically-based ROM is made of the kinematics equation and the law of membrane quasi-static equilibrium in their reduced formulation. The unknown variables become the POD coefficient vectors of displacement and velocity fields. The reduced quasi-static equilibrium law is determined once again by the use of a diffuse approximation. The manifold learning is achieved by the use of time-snapshot data of the interpolated solution at $\theta =\theta_{q}$.

Numerical experiments confirm the efficiency of the method. ROM-vs-FOM speedups are observed to be of order 10,000 with almost the same accuracy (with less than a 0.3% error measured in terms of Hausdorff distance inside the parameter domain). Larger errors are encountered at the boundary of the parameter domain, but they still remain reasonable (up to 3.3% in Hausdorff distance). This work tends to show that model-order reduction techniques are complementary and valuable tools for the rapid design and optimization of capsules in healthcare engineering such as drug delivery through blood vessels.

The case of more complex FSI configurations such as the deformation of capsules flowing through a bifurcated microchannel will be investigated in a future work.

## Figures and Tables

**Figure 1 entropy-23-01193-f001:**
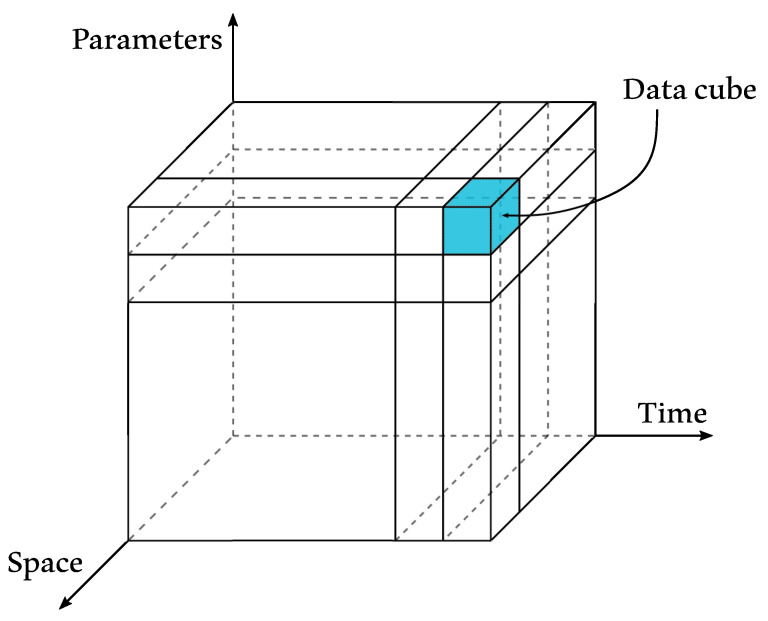
Space-time-parameter data cube.

**Figure 2 entropy-23-01193-f002:**
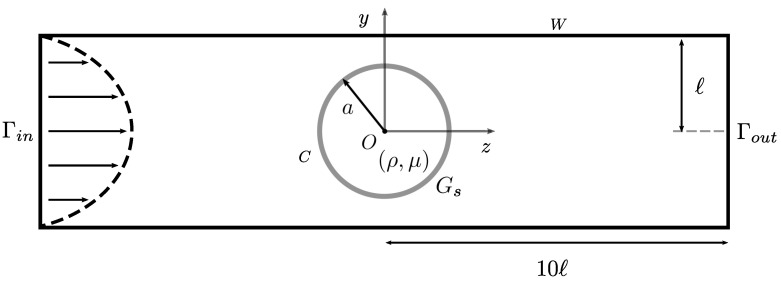
Initial configuration considered in the FOM model: an initially spherical capsule is placed at the center of a square-section channel. The time-evolution of its dynamics is computed using a reference frame centred onto the capsule centre of mass.

**Figure 3 entropy-23-01193-f003:**
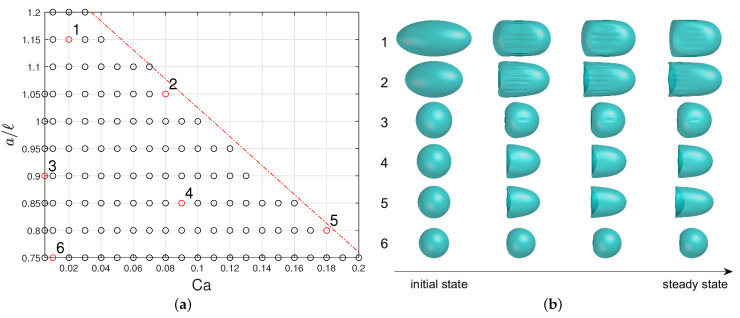
(**a**) Values of Ca and a/l included in the FOM database in the case of an initially spherical capsule with a Neo-Hookean membrane flowing in a square-section microfluidic channel. The parameter domain where a steady capsule deformation can be reached is delimited by the red dotted line. (**b**) Time evolution of capsule deformation along the microfluidic channel shown as illustration for the 6 cases indicated with numbers in figure (**a**). The capsule is pre-deformed into an ellipsoid when a/ℓ≥0.95.

**Figure 4 entropy-23-01193-f004:**
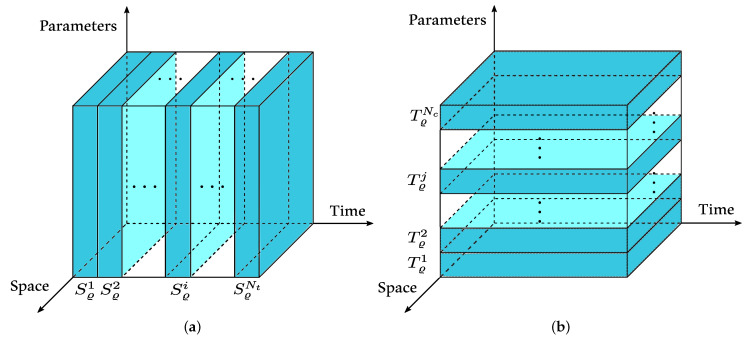
FOM data rearrangements for (**a**) parametric data set selection and (**b**) spatial data set selection.

**Figure 5 entropy-23-01193-f005:**
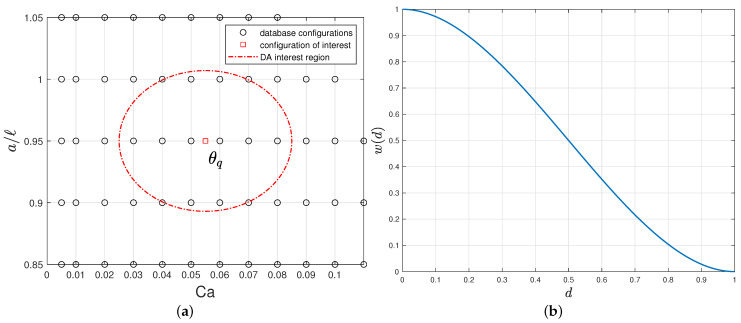
(**a**) DA elliptical region of interest (dashed line) defined around the point $\theta_{q}=\left(Ca=0.055,a/\ell =0.95\right)$ in the parametric space with M=10 neighbors; (**b**) Weight function w(d).

**Figure 6 entropy-23-01193-f006:**
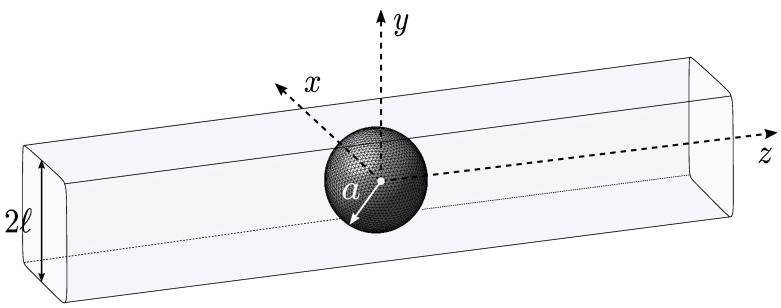
Three-dimensional representation of a capsule flowing in a square microchannel at T=0.

**Figure 7 entropy-23-01193-f007:**
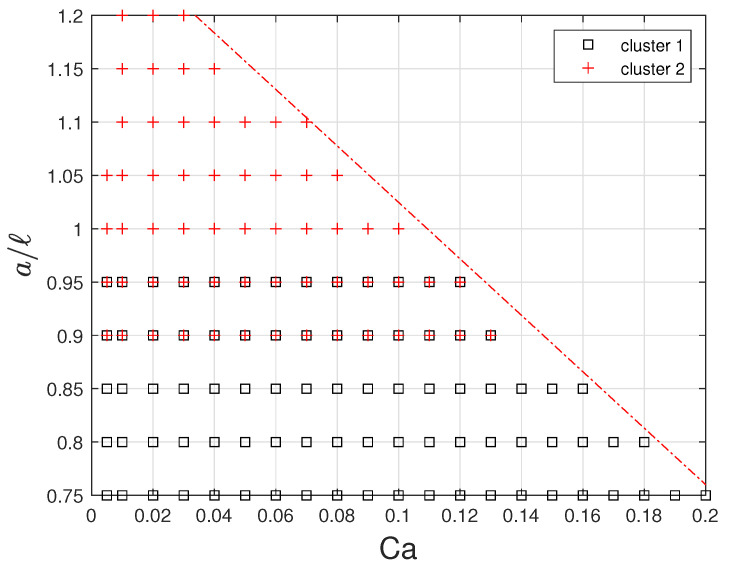
Design of computer experiment with sampling in the admissible parameter domain. The parameter domain is splitted up into two overlapping clusters: cluster 1 (squares), cluster 2 (crosses) and overlapping region (mixed squares and crosses).

**Figure 8 entropy-23-01193-f008:**
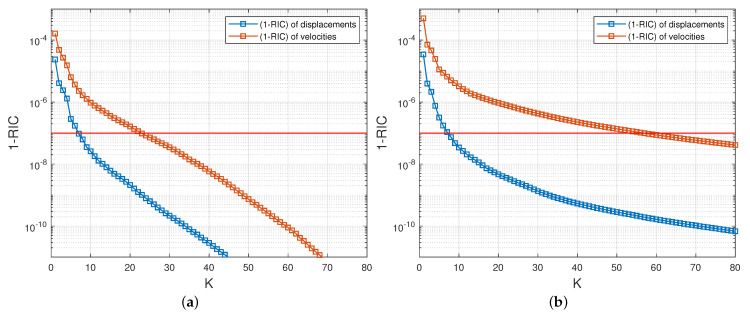
Behaviour of the relative information content of the matrices $S_{u}$ and $S_{v}$ (**a**) and $T_{u}$ and $T_{v}$ (**b**) shown in the form (1−RIC) as a function of the truncation rank *K*. The horizontal red line corresponds to $\left(1-\mathrm{RIC}\right)=10^{-7}$.

**Figure 9 entropy-23-01193-f009:**
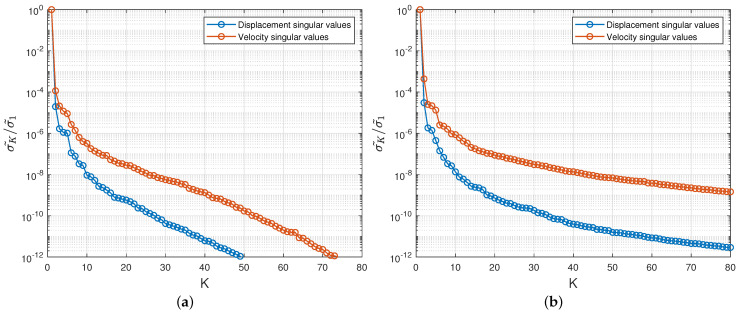
(**a**) Parametric normalized singular values ${\tilde{\sigma }}_{K}/{\tilde{\sigma }}_{1}$ for $S_{u}$ and $S_{v}$; (**b**) Spatial normalized singular values ${\tilde{\sigma }}_{K}/{\tilde{\sigma }}_{1}$ for $T_{u}$ and $T_{v}$.

**Figure 10 entropy-23-01193-f010:**
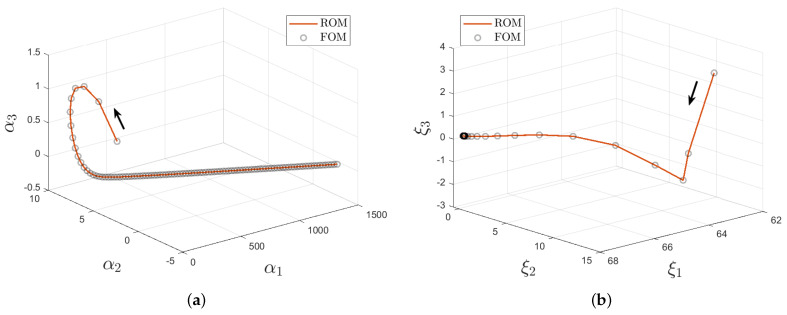
FOM versus ROM comparison of the time evolution of the first three displacement (**a**) and velocity (**b**) POD coefficients for the query parameter $\theta_{q}=\left(0.10,0.90\right)$.

**Figure 11 entropy-23-01193-f011:**
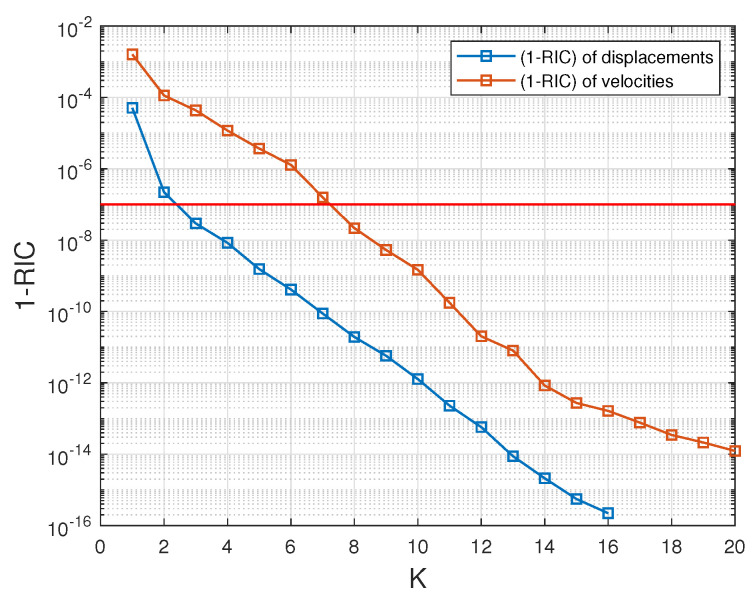
Online stage: behaviour of the relative information content of the matrices $U(\theta_{q})$ and $V(\theta_{q})$ shown in the form (1−RIC) for query parameter $\theta_{q}=\left(0.10,0.90\right)$. The red line corresponds to $\left(1-\mathrm{RIC}\right)=10^{-7}$.

**Figure 12 entropy-23-01193-f012:**
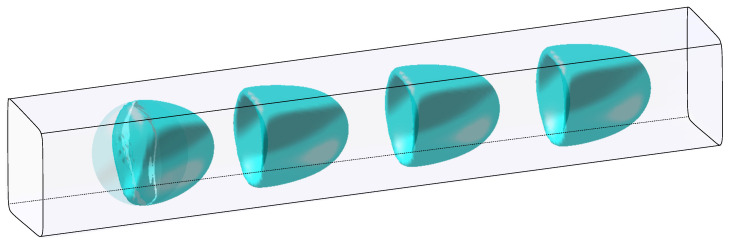
Three-dimensional shape of a capsule flowing in a square microchannel, reconstructed with the ROM model for $\theta =\left(Ca=0.10,a/\ell =0.90\right)$ and shown at $\dot{\gamma }t=0,0.4,3,6,9$. The capsule initial shape is shown in transparency.

**Figure 13 entropy-23-01193-f013:**
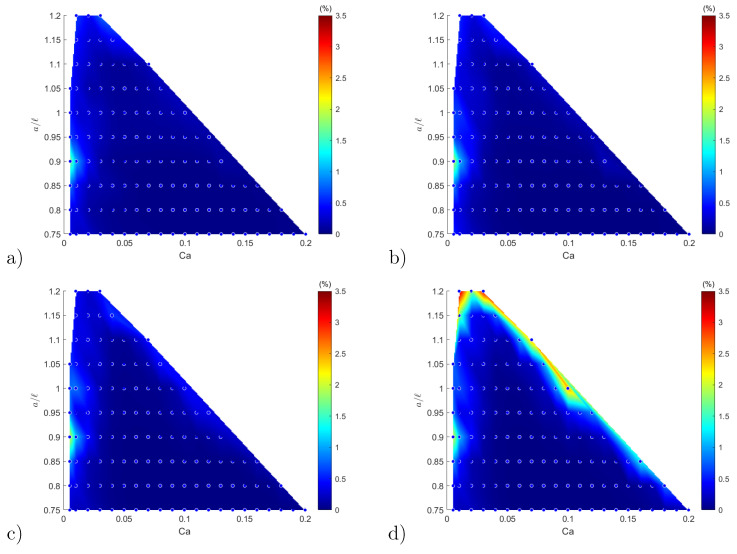
Heat maps of the normalized Hausdorff Distance $d_{MH}/a$ of configuration prediction shapes over the parametric space at different transient states: (**a**) $\dot{\gamma }t=1$; (**b**) $\dot{\gamma }t=2$; (**c**) $\dot{\gamma }t=4$; and (**d**) $\dot{\gamma }t=8$. The maximum error is 3.26% in (**d**).

**Figure 14 entropy-23-01193-f014:**
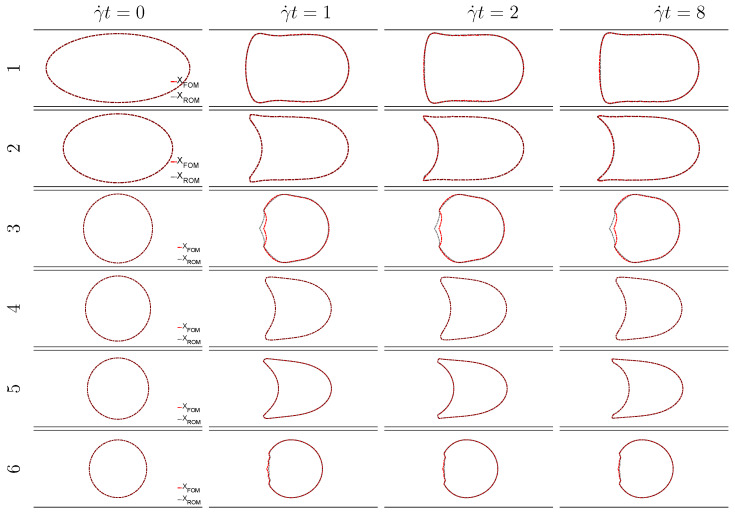
Comparison between the ROM (red dots) and FOM solutions (black line) of the capsule cross-section shapes in the plane y=0 at the times $\dot{\gamma }t=$ 0, 1, 2 and 8 respectively, for the 6 parameter couples selected in Figure 3. The horizontal lines correspond to the channel walls.

**Figure 15 entropy-23-01193-f015:**
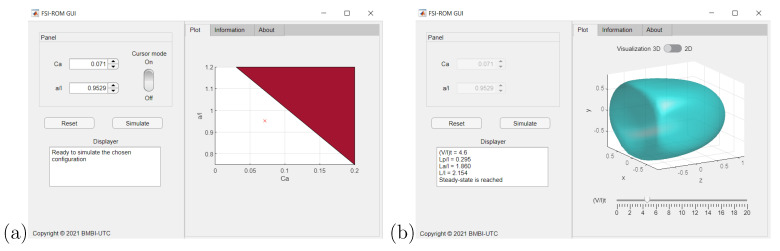
CapsuleExplorer: (**a**) parameter domain exploration; (**b**) dynamic 3D capsule view.

**Figure 16 entropy-23-01193-f016:**
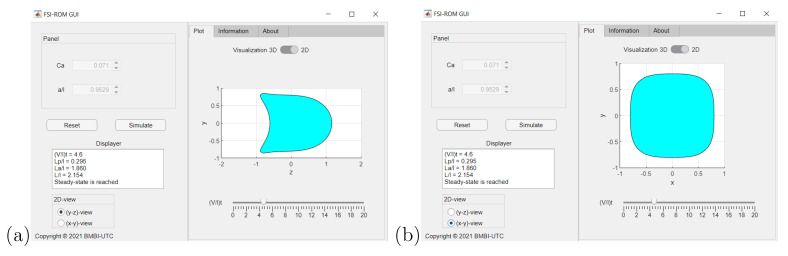
CapsuleExplorer: (**a**) dynamic 2D cross-section longitudinal view; (**b**) dynamic 2D cross-section transversal view.

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
