# Peer review of "A Data-Driven Space-Time-Parameter Reduced-Order Model with Manifold Learning for Coupled Problems: Application to Deformable Capsules Flowing in Microchannels"

_entropy, 2021, doi:10.3390/e23091193_

Round 1

Reviewer 1 Report

In the manuscript under review, the Authors apply a reduced-order modelling method to the problem of hyperelastic capsules moving in a microfluidic channel with square cross section. The flow of the surrounding fluid is in a low-Reynolds-number regime. The data for the full-order solution are generated by coupling boundary element methods for the fluid with finite elements for the capsule surface. This reference data are produced by spanning a given region in the space of parameters (capillary number and particle radius/channel half-width aspect ratio) and for a time window that allows reaching as steady state.

The paper is clearly written, with a detailed methods section, and the results about the reduction method show a good performance of the strategy. The main novelty highlighted in the manuscript is the application to a time-dependent problem. The methods used to tackle time-dependence are similar to those employed for steady problems, but the work is, in my opinion, an interesting addition to the literature about reduced order modelling.

I only have minor concerns about the presentation that the Authors should address before the paper is accepted for publication:

1) In the abstract the application seems to involve "micrometric and nanometric suspensions", suggesting the treatment of a large number of particles, while the paper deals with a single capsule in a micrometric or nanometric channel. This is clearer in the introduction, but the abstract should point more directly to the content of the paper.

2) In equation (8) I would expect to see also the definition of tau_2. Since the membrane is not unstretchable, one cannot deduce tau_2 from tau_1

3) When the aspect ratio is larger than 1.0 there is some arbitrariness in the choice of the initial deformation and contacts between the capsule and the channel wall may appear. Did the Authors check the independence of the steady solution from the initial condition? How were contacts avoided or treated? These points should be addressed in the manuscript.

Reviewer 2 Report

Numerical simulation is essential to study physics-coupled problems such as micrometric suspensions and also for design and optimization of complex systems. The main problem is that the computational time is usually prohibitive. In order to solve that, different kinds of machine learning methods have been used. In this work, a data-driven reduced-order modeling approach is proposed for space-time-parameter mechanical problems. The dynamics of dilute suspensions of micrometric capsules in microfluidic channels is modeled as an application. This is an important example because these microcapsules are used as novel drug carriers.

From the computing performance point of view, impressive speedups are reported, making interactive exploration and real-time visual rendering possible. In fact, a software tool CapsuleExplorer based on the proposed ROM to provide the three-dimensional microcapsule deformation/evolution developed by the authors is shown.

A remarkably clear proposal of a new data-driven reduced-order modeling approach to space-time-parameter mechanical problems shown to be highly efficient, this work certainly merits publication in Entropy.

Reviewer 3 Report

The authors propose a data driven reduced order modeling (ROM) approach for the description of the dynamics of dilute suspensions of micrometric capsules in microfluidic channels. For this a ROM is proposed which contains capsule-fluid interaction which is able to give the time evolution of the capsule shape. The connection between the position and velocity is performed by diffuse approximation and manifold learning using the principal modes of the proper orthogonal decomposition of both position and velocity fields. The manuscript is of interest, but several specifications/extensions must be made in order to reach real publication standards, see below:

1) The use of the Poiseuille flow in the description must be motivated in 
details (why laminar flow ? why incompressible fluid ? why constant pressure
difference drop ?)

2) It should be strongly motivated why the external and internal flow can be
properly described by Stokes equations ?  

3) How influences points 1),2) the a/l ratio at which the provided description 
is valid ?

4) Why square cross section is used for channel ? (It is known that Poiseuille
flow description works usually well for cylindrical tube).

5) What are the velocity limits at which the description is satisfactory ?
(the authors claim that the tub length is ~ 10 l, but in these conditions 
Eq.(2) is not satisfied at high velocity for arbitrary large t).

6) Further explications must be provided why Eq.(9) (the stability condition)
is satisfied. 

7) Fig.3 presents a/l values above 1. Because of the channel wall effects not
considered in the presented theory, in this case the description seems to be
not valid. If yes, motivations must be presented.

8) Eqs.(22,23) practically means separation of variables. Motivations must 
be presented: why such variable separations can be done ?

9) Motivations must be presented for the use of the presented
weight function form from Eq.(27).   

10) Something must be said about the applicability of Eq.(28) (diffuse
approximation) i.e. when is acceptable good, when is misleading ?

11) In Fig.10 the blue line is missing (or cannot be seen), so the FOM to ROM comparison cannot be observed.

12) Since physical phenomena are described, a comparison between theory and experiment is the measure of the "goodness". This is completely missing from the manuscript.

Consequently I cannot recommend the presented version for publication. Instead I suggest an extensive revision on the line presented above.

Round 2

Reviewer 1 Report

The Authors addressed my comments in a satisfactory way.

Author Response

We would like to thank once again Reviewer 1 from constructive comments and proposed corrections.

Reviewer 3 Report

In their response to the referee report, the authors must clearly indicate at each point, the modifications made in the manuscript according to the referee observations made. Furthermore, the authors must provide a list of the modifications made. At the moment I do not see clearly these aspects in the newly provided manuscript. If I receive these required information I can decide about the content of the report. Consequently, at the moment I cannot recommend the paper for publication.

Round 3

Reviewer 3 Report

The authors have reasonably introduced in the text the modifications required by the referee report. There are however three exceptions, which further must be introduced in the text, namely:

a) At point 7): presenting eqs.2-7, the authors should introduce in the text that via these equations the wall effects have been fully considered.

b) At point 9): The response provided at this point must be introduced in the text.

c) At point 12): The response provided at this point must be also introduced in the text.

If these modifications are made, I recommend the paper for publication.
